# Great Diversity of Bacterial Microbiota in Thai Local Food: “*Tai-Pla*”, the Salty Fermented Fish-Entrail Sauce

**DOI:** 10.3390/foods14234104

**Published:** 2025-11-29

**Authors:** Patcharaporn Boonroumkaew, Nongnapas Kanchanangkul, Rutchanee Rodpai, Lakkhana Sadaow, Oranuch Sanpool, Penchom Janwan, Tongjit Thanchomnang, David Blair, Pewpan M. Intapan, Wanchai Maleewong

**Affiliations:** 1Department of Parasitology, Faculty of Medicine, Khon Kaen University, Khon Kaen 40002, Thailand; patcharaporn.bo@kkumail.com (P.B.); nongnapas.kanch@gmail.com (N.K.); rutcro@kku.ac.th (R.R.); sadaow1986@gmail.com (L.S.); oransa@kku.ac.th (O.S.); pewpan@kku.ac.th (P.M.I.); 2Mekong Health Science Research Institute, Khon Kaen University, Khon Kaen 40002, Thailand; penchom.ja@wu.ac.th (P.J.); tongjit.t@msu.ac.th (T.T.); 3Department of Medical Technology, School of Allied Health Sciences, Walailak University, Nakhon Si Thammarat 80161, Thailand; 4Faculty of Medicine, Mahasarakham University, Maha Sarakham 44000, Thailand; 5College of Science and Engineering, James Cook University, Townsville, QLD 4811, Australia; davidblair49@gmail.com

**Keywords:** food fermentation, fermented fish, flavoring food, microbial diversity, microbiota

## Abstract

This study characterized the microbiota by sequencing the V3-V4 regions of prokaryotic 16S rRNA to investigate the bacterial diversity of fermented fish-entrail sauce (*tai-pla* or *pung-pla*) from five provinces in Thailand. *Tai-pla* samples made from seven different species of fish, three freshwater and four marine, were purchased. Three subsamples from each were analyzed. The samples had salt concentrations ranging from 3 to 13% and pH values ranging from 4.26 to 6.19. The top 35 genera of bacterial taxa by relative abundance were considered in more detail. Lactic acid bacteria (LAB), primarily in the order Lactobacillales (*Levilactobacillus*, *Companilactobacillus*, *Lactococcus*, *Latilactobacillus*, *Weissella*, *Pediococcus*, and *Ligilactobacillus*), were abundant in several groups of samples, as were halophilic bacteria, including *Halanaerobium*, *Chromohalobacter*, and *Virgibacillus*. Other beneficial bacterial species were frequently detected, including *Tetragenococcus halophilus* and *Tetragenococcus muriaticus*. Principal Coordinate Analysis visualization of beta diversity showed distinct bacterial community structures across *tai-pla* samples prepared with different fish species. Differences between samples may be due to the use of different raw materials, salt concentrations, recipes, processes and fermentation periods. This study provides baseline information on microbial communities and diversity in *tai-pla*, offering better insights into the production outcomes of traditional products. Further optimization of the fermentation process, such as using beneficial bacterial taxa in starter cultures, may enhance the system of food fermentation, food quality, and flavor control, supporting regulation useful for industrial applications.

## 1. Introduction

Fish-entrail sauce, known in Thailand as “*tai-pla* or *pung-pla*”, is made from raw fish entrails fermented with high amounts of salt. Once properly fermented, it is semi-solid, brown in color, and has a salty taste. Salt is traditionally used for food preservation in Southern Thailand [1,2]. *Tai-pla* is produced by fermenting raw fish entrails with salt in a closed jar and fermenting them for at least 4 weeks. Upon completion, a clear separation between the liquid and solid portions can be observed [1]. The choice of ingredients is an important component, including the selection of fish species and the variety of spices and seasonings that enhance flavor while serving as cultural markers.

*Tai-pla* is used as seasoning in various dishes, such as *kaeng tai-pla*, a traditional curry in Southern Thailand, and as a dipping sauce in Central Thailand. *Kaeng tai-pla*, a unique Southern Thai curry cuisine, is popular in every region of Thailand due to its strong taste, spicy flavor, and unique aroma [2]. Fermentation involves interactions between enzymes and microorganisms that decompose the raw materials (proteins, fats, and carbohydrates). This process can be started naturally (by ambient bacterial species) or by adding starter enzymes or microorganisms; for example, *Bacillus subtilis* has been used for the fermentation of scales and intestines, and *Lactobacillus pentosus* has been applied to catfish and tilapia fish waste [3]. Use of a starter culture also plays a role in standardizing quality and accelerating fermentation [4].

The microbial diversity of various fermented foods, such as Chinese fermented mandarin fish (*Siniperca chuatsi* or *chouguiyu*) [5,6] and Thai fermented fish (*pla-ra*) [7], has recently been investigated using next-generation sequencing. Certain microorganisms in fermented fish, especially lactic acid bacteria (LAB), enhance flavor, ensure safety, and outcompete spoilage organisms while limiting the production of harmful biogenic amines during fermentation [8].

There has been no previous report on the bacterial diversity of *tai-pla* in Thailand based on metagenomics. Thus, we aimed to study the bacterial diversity in *tai-pla* samples fermented from seven different types of freshwater or marine fish. To achieve this, we used sequences from the 16S rRNA (V3-V4 regions) generated using Illumina Nova sequencing. This study may provide basic knowledge to guide the selection of suitable bacteria for the production of *tai-pla* products, potentially increasing their value in the future.

## 2. Materials and Methods

### 2.1. Sample Collection and Physicochemical Analysis

Salty fermented fish-entrail sauces (*tai-pla* or *pung-pla*) from five provinces in Thailand, each made exclusively from a single fish species, were purchased from local markets between February 2021 and August 2023 (Table 1). Three subsamples were analyzed from each. Groups (each consisting of three subsamples) 1, 4 and 6 were based on freshwater fish and the remainder employed marine fish as follows: Group 1, *Trichopodus* sp.; Group 2, *Rastrelliger* sp.; Group 3, *Mugil* sp.; Group 4, *Oreochromis* sp.; Group 5, *Euthynnus* sp.; Group 6, *Channa* sp.; and Group 7, *Priacanthus* sp. After purchase, the samples were placed in foam boxes for transport to the laboratory. Aliquots for DNA extraction were frozen at −20 °C to minimize further microbial activity prior to use. A ten-fold dilution of each subsample was prepared using deionized water to determine the salt concentration, which was measured directly using a salt meter (DRETEC Co., Ltd., Saitama, Japan). The pH of each *tai-pla* sample was directly measured with a pH meter (Mettler Toledo, Columbus, OH, USA). These measurements were performed on the day when the subsample was first thawed. The physicochemical analyses were repeated three times for each group (once per subsample) and are reported as the mean and standard deviation.

### 2.2. Extraction of DNA from Tai-Pla

A portion (200 mg) of each *tai-pla* subsample was used for total DNA extraction with the QIAamp^®^ Fast DNA stool mini kit (Qiagen, Hilden, Germany), following the manufacturer’s instructions. DNA concentration and purity were measured using a NanoDrop™ One Spectrophotometer (Thermo Fisher Scientific, Waltham, MA, USA) and via 1% agarose electrophoresis. Subsequently, each DNA sample was diluted to a final concentration of 1 ng/µL with sterile water [7].

### 2.3. Bacterial 16S rRNA Gene Amplification and Sequencing

DNA from 21 samples was used for amplification with Phusion^®^ High-Fidelity PCR Master Mix (New England Biolabs, Inc., Ipswich, MA, USA) and specific primers 341F (5′-CCT AYG GGR BGC ASC AG-3′) and 806R (5′-GGA CTA CNN GGG TAT CTA AT-3′) that amplify the V3−V4 (barcode) regions of the 16S rRNA gene (NovogeneAIT Genomics, Singapore, Singapore). The thermal cycling conditions consisted of initial denaturation at 98 °C for 1 min, followed by 30 cycles of denaturation at 98 °C for 10 s, annealing at 50 °C for 30 s, and elongation at 72 °C for 30 s and 72 °C for 5 min. PCR products of the expected size (400–450 bp in length) were purified and used for library preparation (NovogeneAIT Genomics).

Libraries were sequenced on a paired-end Illumina platform to generate 250 bp paired-end raw reads. Subsequently, library quality was checked with Qubit and quantified via real-time PCR and a bioanalyzer to assess size distribution. Quantified libraries were pooled and sequenced on Illumina platforms (NovogeneAIT Genomics).

### 2.4. Bioinformatics and Statistical Analyses

Sequences were processed using the QIIME2 pipeline. The paired-end reads were assigned to samples depending on their unique barcodes, with barcode and primer sequences removed, assembled using FLASH (Version 1.2.11) [9]. Raw tags were filtered using fastp software (Version 0.23.1) to obtain high-quality clean tags [10]. The Silva database (release 138.1) was used to identify chimeric sequences, which were removed using the Vsearch package (V2.16.0) [11].

The remaining (effective) tags were denoised using the DADA2 plugin of QIIME2 (version 2022.02) to generate initial amplicon sequence variants (ASVs). Each ASV was compared to the Silva database using the classify-sklearn algorithm in QIIME2 software to obtain the taxonomy annotation and the abundance of each taxon at the of kingdom, phylum, class, order, family, genus, and species levels. Histograms of relative abundance of the top 10 taxa at phylum, family, genus, and species levels were plotted in perl with the SVG function. A heatmap of the top 35 genus-level taxa was constructed in R using the pheatmap function. Venn and flower diagrams were produced in R using the Venn diagram function and in perl with the SVG function, respectively.

Alpha diversity was calculated in QIIME2 from richness (observed species and Chao1) and diversity (Shannon and Simpson) indices. Statistically significant differences between groups were assessed using the Kruskal–Wallis pairwise test. The abundance of different ASVs was used to generate a rarefaction curve for estimating the richness and diversity in the microbiota of the seven *tai-pla* sample groups. Beta diversity was assessed using Principal Coordinates Analysis (PCoA) based on Jaccard distance, which measures the differences between groups.

### 2.5. Accession Numbers

All sequence reads have been deposited in the NCBI Sequence Read Archive (SRA) under project accession number PRJNA1356154.

## 3. Results

### 3.1. Physicochemical Properties and Microbial Taxonomic Composition in Tai-Pla Samples

The pH ranged from slightly acidic to near-neutral (4.26 to 6.19), while the salinity, expressed as the amount of NaCl, ranged from 3 to 13% (*w*/*v*) (Table 1). Basic statistics of the number of reads and ASVs in each sample are shown in Appendix A. A total of 4,092,419 effective tags were obtained, with the number per subsample ranging from 177,689 to 207,735 (Appendix A). The complete list of bacteria found includes 4350 ASVs (Appendix A) in 48 phyla, 108 classes, 222 orders, 317 families, 622 genera, and 316 species (Appendix A). Data from all three subsamples per group were combined by group in most of the analyses that follow.

Bacillota was the most-represented phylum in four groups (Groups 5 (97.2%), 4 (91.6%), 7 (78.3%), and 1 (46.3%)). However, Pseudomonadota was abundant in all groups, especially in Groups 2 (72.8%) and 6 (89.9%). Halanaerobiaeota was abundant in Groups 3 (82.1%) and 7 (16.2%). The remaining seven of the top ten phyla showed relative abundances ranging from 8.9% to less than 1% (Appendix A). The top 10 families varied among sample groups: Enterococcaceae were abundant in Groups 5 and 7, Lactobacillaceae was abundant in Group 4, Halanaerobiaceae were abundant in Group 3, and Lysobacteraceae were abundant in Groups 1, 2 and 6. The remaining six families accounted for relative abundances ranging from 19.3% to less than 1% (Figure 1A).

At the genus level, *Tetragenococcus* was most abundant in Groups 7 (75.0%) and 5 (70.9%). *Halanaerobium* was most abundant in Group 3 (82.3%). *Stenotrophomonas* was most abundant in Groups 6 (72.7%), 2 (51.9%) and 1 (29.0%). *Latilactobacillus* was most abundant in Group 4 (59.3%) (Figure 1B). At the species level, *Tetragenococcus halophilus* was most abundant in four groups, and in Group 3, it was co-abundant with *Lentibacillus juripiscarius*. Meanwhile, *Lactobacillus namurensis* was abundant in Group 4. The other six species ranged in abundance from 13.8% to less than 1% of reads (Appendix A). The heatmap showing the most abundant 35 genera (Figure 2) reveal variation among groups. Lactic acid bacteria (LAB) with potential probiotic properties were abundant in some groups: *Levilactobacillus*, *Companilactobacillus*, *Lactococcus*, *Latilactobacillus*, and *Weissella* in Group 4, *Pediococcus* and *Ligilactobacillus* in Group 5, and *Lentibacillus* in Group 7. In contrast, pathogenic bacteria were abundant in Group 1 (*Bacteroides* and *Paeniclostridium*) and Group 4 (*Enterobacter* and *Klebsiella*). Halophilic bacteria, including *Halanaerobium*, *Chromohalobacter*, and *Virgibacillus*, were abundant in Group 3, and *Synechococcus* CC9902 in Group 7 (made from a marine fish). The genus *Akkermansia* was abundant in Group 1 and *Delftia* and *Chryseobacterium* occupied a similar position in Group 6.

### 3.2. The Number of Shared and Unique Bacterial ASVs Among Tai-Pla Sample Groups

A flower diagram analysis revealed only one core (shared) ASV present in all sample groups (Figure 3A and Appendix A). ASVs represented the sequence compared with 99%. Group 1 had the most ASVs. All groups had one or more unique ASVs: Groups 1 (2170 unique ASVs), 2 (271 ASVs), 3 (205 ASVs), 4 (432 ASVs), 5 (272 ASVs), 6 (384 ASVs), and 7 (181 ASVs) (Figure 3A and Appendix A). Venn diagrams separate out data from *tai-pla* samples from freshwater fish groups (Figure 3B and Appendix A) and marine fish groups (Figure 3C and Appendix A).

### 3.3. Bacterial Community Diversity and Similarity

Bacterial community richness and diversity in each group were assessed using alpha diversity metrics. Alpha rarefaction analysis was used to determine whether sufficient sampling effort had been carried out (Appendix A). Alpha diversity indices revealed significant differences between Group 1 and other groups in Chao1 indices and observed richness (Figure 4A). The Shannon and Simpson metrics for diversity also show significant differences between groups (Figure 4B).

Beta diversity of bacterial communities within and among groups was assessed using the Jaccard distance and visualized by means of PCoA (Figure 5). The PCoA plot revealed clear differences in bacterial community structure among *tai-pla* samples prepared with different fish species. The plot also grouped subsamples together for all groups, except Groups 1 and 2—in these, the subsamples were placed far apart, indicating different microbial communities in each subsample.

## 4. Discussion

We used next-generation sequencing technology, the 16S rRNA Illumina Nova sequencing (V3-V4 regions) platform, for a comprehensive study of bacterial communities in *tai-pla* purchased in five provinces of Thailand and made from seven different fish species. Physicochemical analysis revealed considerable variation among groups, with pH levels ranging from 4.26 to 6.19 and salinity concentrations varying from 3 to 13% (*w*/*v*) (Appendix A). By comparison, a previous report on *tai-pla* documented salt contents of 13.5–25.3% (*w*/*v*) [1]. Salt is one of the most extensively utilized raw materials in fermented fish production. Indeed, fermented fish products are primarily categorized according to salt concentration. High-salt formulations contain more than 20% salt by weight, while in medium-salt products, salt concentrations range from 10% to 15%. Low-salt variants are characterized by salt content between 3% and 8%, and salt-free products constitute a specialized category [8]. The starting salt concentration may be one of the factors that dictates levels of lactic acid bacteria growth and their products, which in turn affect other parameters in *tai-pla*. This is to be expected given the absence of standardized formulations and preparation protocols for these traditional products.

The phylum Halanaerobiaeota (represented by the halophilic family Halanaerobiaceae and genus *Halanoaerobium*) was abundant in Group 3 (82.1%), which was characterized by a salinity of 13% (*w*/*v*), the highest among our samples (Appendix A). The *tai-pla* in Group 3 was prepared from marine *Mugil* spp. *Halanaerobium praevalens*, the type species of *Halanaerobium*, is of particular interest due to its capacity to reduce a wide range of nitro-substituted aromatic compounds at a high rate. Furthermore, it is capable of degrading organic pollutants, a function relevant in ecosystems shared with sulfate-reducing and methanogenic bacteria [12]. Other genera of halophilic bacteria (*Chromohalobacter* and *Virgibacillus*) were also abundant in Group 3. These are found in a variety of fermented foods, such as fish sauce, soy sauce, and shrimp paste [13,14,15].

Phylum Bacillota was highly represented in all *tai-pla* samples and the most abundant phylum, except there were a few in Groups 3 and 6. (Appendix A). However, Pseudomonadota was also abundant in all sample groups. Similar findings have been reported in traditional fermented fish (*budu*) from Malaysia and Indonesia [16,17] and traditional fermented fish (*pla-ra*) from Thailand [7]. A healthy gut microbiota is abundantly composed of phyla Bacillota and Bacteroidota [18].

The Lactobacillaceae (Phylum Bacillota), a diverse family of LAB from the gut microbiota of humans and animals, confers key benefits to host health. These effects include immunomodulation and pathogen inhibition, properties that are the reason for the widespread use of many species as probiotics [19]. In this study, members of Lactobacillaceae were abundant in Group 4, corresponding to *tai-pla* prepared from *Oreochromis* sp., a freshwater fish, which showed the lowest salinity level (3%) in the fermentation process. The Enterococcaceae (Bacillota) was detected in nearly all groups but was abundant in Group 5 and Group 7, corresponding to *tai-pla* prepared from the marine *Euthynnus* sp. and *Priacanthus* sp., with fermentation salinity levels of 9% and 11%, respectively. Probiotics derived from this family, particularly *Enterococcus* species, are employed in the management of irritable bowel syndrome, infectious diarrhea, and antibiotic-associated diarrhea. Furthermore, these microorganisms have been shown to reduce cholesterol levels and enhance host immunity [20].

Genera of LAB were abundant in Group 4 (*Levilactobacillus*, *Companilactobacillus*, *Lactococcus*, *Latilactobacillus*, and *Weissella*), Group 5 (*Pediococcus* and *Ligilactobacillus*), and Group 7 (*Lentibacillus*) and possess potential probiotic properties. LAB enhance the production of metabolites that contribute to flavor complexity, which is crucial for flavor development and food preservation, as these bacteria also produce antimicrobial substances that prevent the spread of pathogenic and spoilage microorganisms [3,8].

At the species level (Appendix A), the LAB *Tetragenococcus halophilus* was the most abundant species in Groups 5, 7, 2, and 6, which corresponded to *tai-pla* samples fermented at salinity levels ranging from 9% to 11%. This species produces lactic and short-chain acids that enhance the product’s flavor and lower its pH [21]. In addition, for control of harmful taxa, the incorporation of sucrose and *T. halophilus* during the mixing stage can accelerate pH decline in fish-sauce mashes, restraining histamine-producing bacteria and improving final amino acid profiles [22]. *Tetragenococcus muriaticus* was abundant in *tai-pla* prepared from marine fish (Groups 2 and 7) and can produce halophilic glutaminase, an enzyme that enhances the umami flavor in foods [23]. This species is also abundant in fermented products such as grasshopper sub shrimp paste, where it helps reduce biogenic amines, improving flavor and safety [24].

Some LAB strains, such as *Lactobacillus namurensis* NH2 and *Pediococcus pentosaceus* HN8 (neither of which we found), can produce γ-aminobutyric acid (GABA) and have been used as starter cultures for *nham* (Thai fermented pork sausage) [25]. Moreover, *Lactobacillus acidophilus*, which was most abundant in *tai-pla* prepared from *Euthynnus* sp. (Group 5), functions as an effective adjunct culture in food fermentation processes. This probiotic bacterium contributes to the development of distinctive taste profiles, complex flavor notes, and improved textural properties in fermented food products. It also preserves the products by producing lactic acid and bacteriocins (which produce antimicrobial compounds) and metabolites to decrease the growth of human pathogens [26]. Similarly, *Weissella jogaejeotgali* (Bacillota), originally isolated from *jogae jeotgal*, a traditional Korean fermented seafood [27], was also found to be abundant in *tai-pla* from *Euthynnus* sp., suggesting that multiple LAB species contribute to the flavor development and safety of this fermented fish product. Additionally, two halophilic LAB species are notable for their remarkable salt tolerance and adaptability. *Lentibacillus kimchii* (Bacillota) isolated from *kimchi*, a traditional Korean fermented vegetable product, can grow in 10.0–30.0% (*w*/*v*) NaCl (optimum, 15.0–20.0%), pH 7.0–8.0 (optimum, pH 7.5), and temperatures of 15–40 °C (optimum, 30 °C) [28]. Similarly, *Lentibacillus juripiscarius*, isolated from Thai fish sauce (*nam-pla*), is capable of growing under a wide range of conditions, including temperatures of 10–45 °C, salinity of 3–30% NaCl, and pH values of 5–9 [29]. In our study, both species were detected, with *L. kimchii* being most abundant in *tai-pla* prepared from *Priacanthus* sp. (Group 7) and *L. juripiscarius* being most abundant in *tai-pla* from *Mugil* sp. (Group 3).

Members of other phyla represented in our samples can also have beneficial effects. For example, Pseudomonadota (proteobacteria) produces a class of bioactive peptides through nonribosomal pathways, which have demonstrated potent antibacterial and/or antifungal properties [30]. Members of the phylum Actinomycetota, family Micrococcaceae, play a key role in the synthesis of bioactive compounds exhibiting a wide spectrum of therapeutic and antimicrobial properties [31].

Although some strains at a general level are potentially pathogenic, some species have benefits. For example, *Klebsiella pneumoniae* (Pseudomonadota) has demonstrated enhanced production of D-lactic acid [32]. *Synechococcus* CC9902 (Cyanobacteriota) was abundant in Group 7, made from a marine fish. *Synechococcus* is widely distributed in various marine environments and has an important role in carbon export processes [33]. *Akkermansia* (Verrucomicrobiota), abundant in Group 1, has potential probiotic properties and promotes gut health [34]. *Delftia* (Pseudomonadota) and *Chryseobacterium* (Bacteroidota) were abundant in Group 6. *Delftia* is part of the microbial community producing specific flavor compounds during the fermentation of rice wine [35].

Interestingly, Yongsmith and Malaphan (2016) [1] reported *Bacillus polymyxa*, *Bacillus subtilis*, *Pediococcus halophilus*, *Staphylococcus aureus*, *Staphylococcus epidermidis*, and *Vibrio fischeri* from *tai-pla*. These species were not detected in our study. These bacteria include both beneficial species, such as *Bacillus* spp. and *Pediococcus halophilus*, which can enhance flavor, produce enzymes, and support fermentation, as well as potentially harmful species, such as *Staphylococcus aureus* and *Vibrio fischeri*, which may pose food safety risks.

Many bacterial species were found in the *tai-pla* samples made from different types of fish. The considerable taxonomic diversity in the microbiota among these samples may be due to the different raw materials used, salt concentrations, recipes, processes, and fermentation periods [7,17]. Knowledge of the microbial communities in *tai-pla* samples provides a better understanding of these traditional products. Our new understanding of bacterial species found in *tai-pla* highlights the diversity of bacteria that may be beneficial to health or enhance flavor and seasoning. However, we also found some unclassified ASVs. The role of these in initiating fermentation is still unknown, as are their possible beneficial or harmful effects.

Readers should note that we used commercially prepared samples of *tai-pla*. No information was available to us about the exact recipe used, whether a starter culture was employed, or the length of the fermentation period before our purchase. Nevertheless, our findings from a range of *tai-pla* samples should point towards possible optimization of the fermentation process, using some dominant bacterial taxa in starter cultures. This may help improve processes of food fermentation, food quality, and flavor control, providing beneficial outcomes for industrial applications.

## 5. Conclusions

Metagenomic analysis of the salty fermented fish-entrail sauce “*tai-pla*” from freshwater and marine fish revealed great diversity in bacterial microbiota. Lactic acid bacteria, including *Companilactobacillus*, *Lactococcus*, *Latilactobacillus*, *Weissella*, *Pediococcus*, and *Ligilactobacillus*, were abundant across several sample groups. Halophilic bacteria, including *Halanaerobium*, *Chromohalobacter*, and *Virgibacillus*, were abundant, especially in samples made from marine fish. Other beneficial bacterial species that enhance umami flavor, including *Tetragenococcus halophilus* and *Tetragenococcus muriaticus*, were commonly detected. Notably, pathogenic bacteria were also abundant in some groups. Knowledge of the microbial communities in *tai-pla* products provides a better understanding of these traditional products and reveals new insights into the classification of bacterial species found in *tai-pla*, as well as the diversity of bacteria that may benefit health or enhance flavor and seasoning. However, the discovery of various harmful bacterial species indicates that further development is needed in the production process to improve product quality for industrial applications.

## Figures and Tables

**Figure 1 foods-14-04104-f001:**
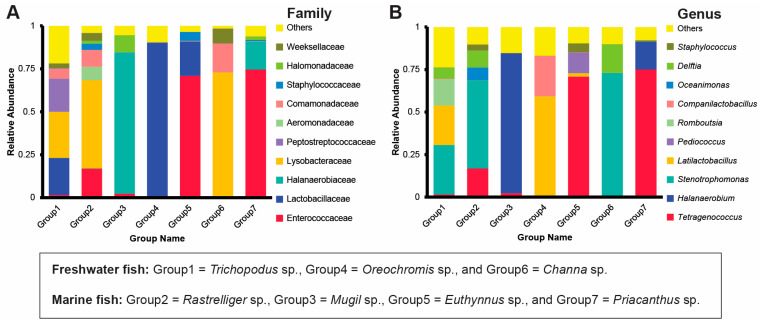
The taxonomic composition and relative abundance of microbial communities in *tai-pla* samples. The bar graphs represent the top 10 families (**A**) and genera (**B**).

**Figure 2 foods-14-04104-f002:**
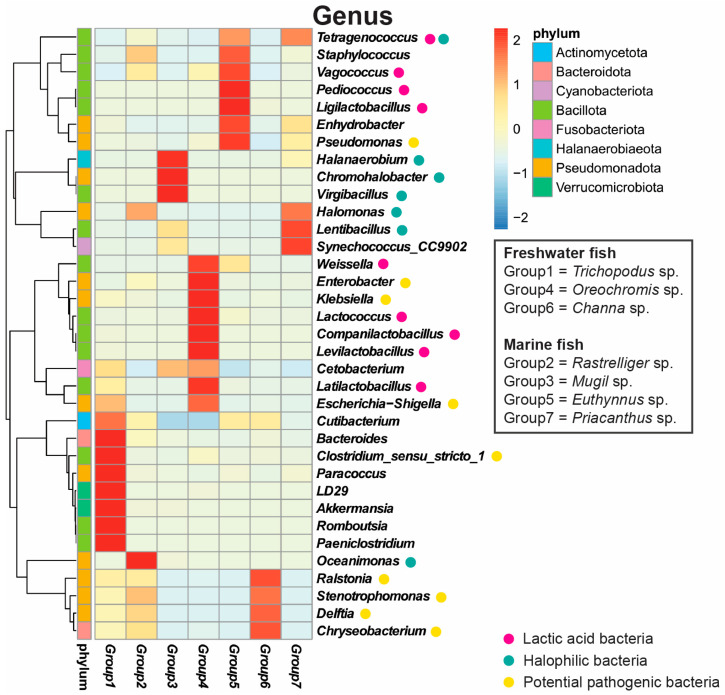
The heatmap represents the top 35 genera; colors indicate the relative abundance of each ASV in each sample group category plotted by the absolute z-score.

**Figure 3 foods-14-04104-f003:**
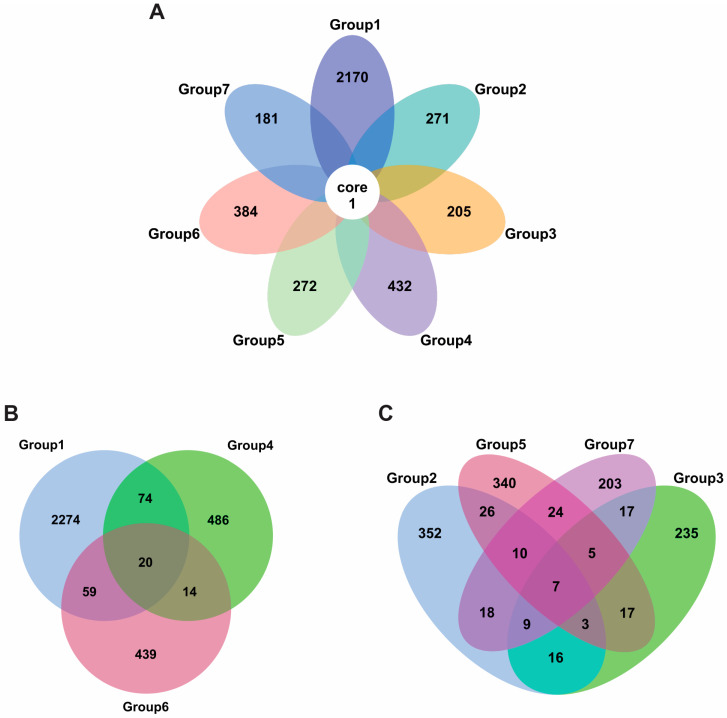
Flower and Venn diagrams showing the number of shared ASVs and unique ASVs in the *tai-pla* samples. Each petal in the flower diagram (**A**) represents one group. The number in the white circle indicates the sole bacterial ASV found in all groups. The numbers in each petal indicate the number of bacterial ASVs unique to that group. Venn diagrams showing shared and unique ASVs from *tai-pla* made from freshwater fish species (**B**) and marine fish (**C**). Group 1 = *Trichopodus* sp., Group 2 = *Rastrelliger* sp., Group 3 = *Mugil* sp., Group 4 = *Oreochromis* sp., Group 5 = *Euthynnus* sp., Group 6 = *Channa* sp., and Group 7 = *Priacanthus* sp.

**Figure 4 foods-14-04104-f004:**
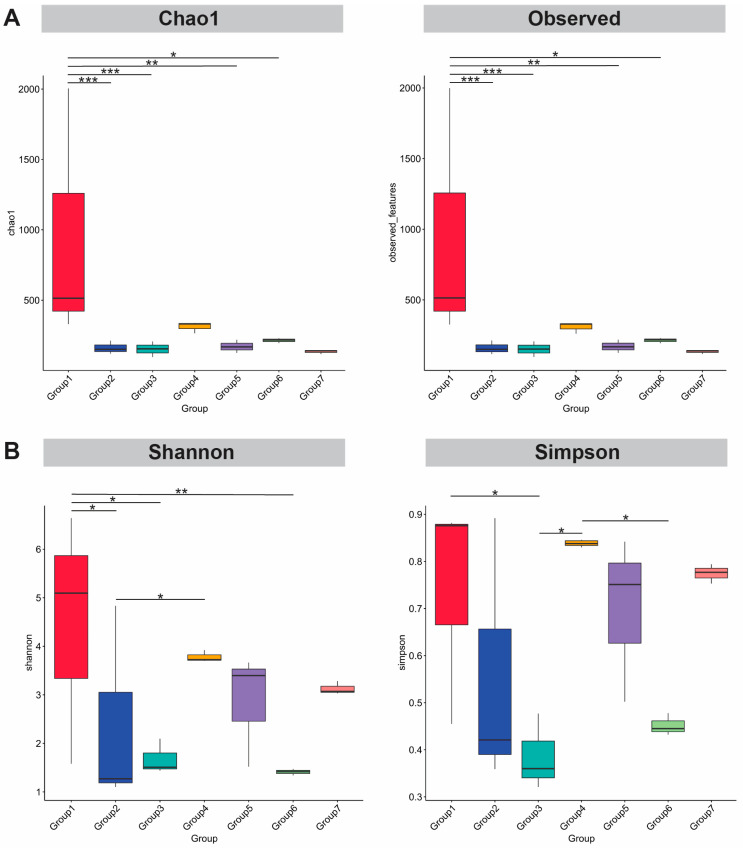
Bacterial community diversity between groups of *tai-pla*. Boxplot of alpha diversity indices. The Chao 1 and observed indices estimate the ASV richness in *tai-pla* samples (**A**). The Shannon and Simpson metrics for ASV diversity in *tai-pla* samples (**B**). The Kruskal–Wallis pairwise test was used to assign statistically significant differences between groups (* *p* < 0.05, ** *p* < 0.01, *** *p* < 0.001). Group 1 = *Trichopodus* sp., Group 2 = *Rastrelliger* sp., Group 3 = *Mugil* sp., Group 4 = *Oreochromis* sp., Group 5 = *Euthynnus* sp., Group 6 = *Channa* sp., and Group 7 = *Priacanthus* sp.

**Figure 5 foods-14-04104-f005:**
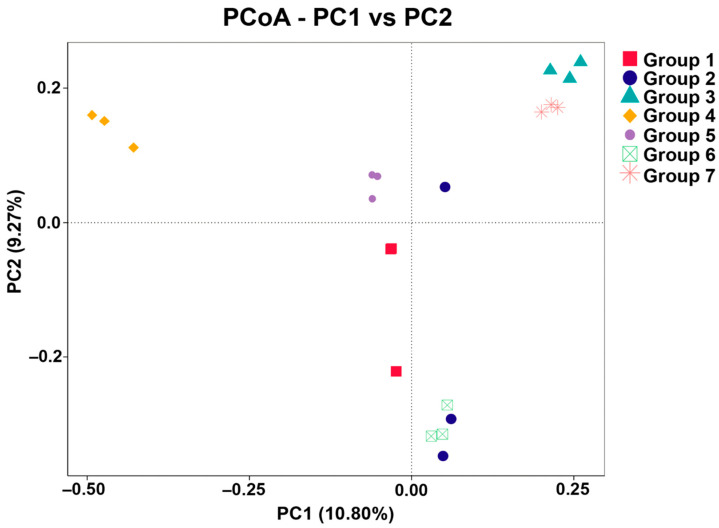
PCoA plot showing the differences in bacterial community structure among the *tai-pla* subsamples within and among groups. Group 1 = *Trichopodus* sp., Group 2 = *Rastrelliger* sp., Group 3 = *Mugil* sp., Group 4 = *Oreochromis* sp., Group 5 = *Euthynnus* sp., Group 6 = *Channa* sp., and Group 7 = *Priacanthus* sp.

**Table 1 foods-14-04104-t001:** Types of fish, production region, salinity, and pH values of *tai-pla* products analyzed in this study.

Group	Type of Fish	Location of Sample Group	pH ± SD	Salinity (%*w*/*v*)
1	*Trichopodus* sp.	Pak Phanang, Nakhon Si Thammarat	4.26 ± 0.01	7
2	*Rastrelliger* sp.	Pak Phanang, Nakhon Si Thammarat	5.35 ± 0.00	11
3	*Mugil* sp.	Mueang Khon Kaen, Khon Kaen	5.74 ± 0.02	13
4	*Oreochromis* sp.	Nam Phong, Khon Kaen	5.02 ± 0.02	3
5	*Euthynnus* sp.	Takua Pa, Phang Nga	4.70 ± 0.22	9
6	*Channa* sp.	Wang Noi, Ayutthaya	5.04 ± 0.01	11
7	*Priacanthus* sp.	Ao Luek, Krabi	6.19 ± 0.07	11

## Data Availability

The original contributions presented in the study are included in the willarticle; further inquiries can be directed to the corresponding author.

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
