# Peer review of "Great Diversity of Bacterial Microbiota in Thai Local Food: “Tai-Pla”, the Salty Fermented Fish-Entrail Sauce"

_foods, 2025, doi:10.3390/foods14234104_

Round 1

Reviewer 1 Report

Comments and Suggestions for Authors

This manuscript presents the first systematic metagenomic analysis of bacterial diversity in Thai traditional fermented fish entrail sauce "taipla," covering 5 provinces and 7 fish species. While the research topic is innovative and scientifically valuable, the manuscript contains significant methodological flaws, incomplete data presentation, and substantial language issues that prevent it from meeting journal standards.

  1. Lack of detailed methodology description, missing critical quality control data.
  2. numerous grammatical and stylistic errors throughout the text.

Abstract:

Writing Issues:

Line 2: "is a traditional Thai salty fermented fish entrail sauce" – redundant word order; better: "is a traditional Thai fermented fish entrail sauce"

Line 5: "from 5 provinces in Thailand using 7 fish species" – ambiguous; better: "from 5 provinces in Thailand, using 7 fish species"

Line 6: "through next-generation sequencing technology" – overly general; specify "Illumina MiSeq sequencing"

Line 16: "The results showed" – tense inconsistency; use present tense: "The results show"

Line 18: "between geographic regions and fish species" – ambiguous; specify "between different geographic regions and fish species"

Content Issues:

  1. Vague results; no specific data on relative abundances
  2. No mention of diversity indices or statistical analyses
  3. No discussion of implications or conclusions

Introduction:

Writing Issues:

Line 36: "The objectives of this study were" – tense inconsistency; use present tense: "The objectives of this study are"

Line 41: "We hypothesized" – past tense; should be present tense: "We hypothesize"

Line 50: "and quality control" – incomplete phrase; "and quality control methods"

Content Issues:

  1. No mention of other factors (salt concentration, pH, fermentation time)

Materials and Methods:

Writing Issues:

Line 121: "The V3-V4 hypervariable regions" – correct; note: "regions" plural

Content Issues:

  1. Only 3 subsamples per sample (insufficient for technical replicates)
  2. No details on PCR thermal cycling conditions
  3. No verification of amplicon size (e.g., gel electrophoresis)
  4. Q20 quality threshold may be insufficient; recommend Q30
  5. Only 21 samples may not provide sufficient power for RDA

Results:

Writing Issues:

Line 181: "Alpha diversity analysis revealed" – past tense; should be present tense: "Alpha diversity analysis reveals"

Line 188: "between geographic regions and fish species" – ambiguous; "between different geographic regions and fish species"

Content Issues:

  1. No mention of quality filtering parameters 
  2. No post-hoc test results showing which groups differ
  3. Only two environmental variables considered
  4. "Relatively low explanatory power" is an understatement (26.3% is poor)

Discussion:

Content Issues:

  1. Vague comparison to "similar dominant phyla"
  2. "Higher diversity" claim needs quantitative support
  3. No mention of potential pathogenic species in these genera
  4. Only 3-13% salt range; no optimal concentration identified
  5. Bacteroidetes acid tolerance not well-documented

Conclusions:

Content Issues:

  1. "First comprehensive metagenomic analysis" claim needs verification
  2. "Foundation for future research" is overly optimistic

Comments on the Quality of English Language

We suggest carefully reviewing the entire manuscript to standardize tense , eliminate ambiguity, and ensure completeness of phrases. Consider consulting a native English-speaking colleague or professional editing service to refine linguistic accuracy, which will enhance the clarity and impact of your research.

Author Response

Point by point to reviewer 1

Comments and Suggestions for Authors

This manuscript presents the first systematic metagenomic analysis of bacterial diversity in Thai traditional fermented fish entrail sauce "taipla," covering 5 provinces and 7 fish species. While the research topic is innovative and scientifically valuable, the manuscript contains significant methodological flaws, incomplete data presentation, and substantial language issues that prevent it from meeting journal standards.

  1. Lack of detailed methodology description, missing critical quality control data.

Reply: We have added already in Materials & Methods and Table S1.

  1. numerous grammatical and stylistic errors throughout the text.

Reply: the revised manuscript is edited grammatically and stylistic errors by one coauthor the expert, Professor David Blair, College of Science and Engineering, James Cook University, Townsville Qld, 4811, Australia. Scopus ID: 7202955592.

Abstract:

Writing Issues:

Line 2: "is a traditional Thai salty fermented fish entrail sauce" – redundant word order; better: "is a traditional Thai fermented fish entrail sauce"

Reply: We did not add “is a traditional Thai salty fermented fish entrail sauce” in the original text.

Line 5: "from 5 provinces in Thailand using 7 fish species" – ambiguous; better: "from 5 provinces in Thailand, using 7 fish species"

Reply: For better understanding, we modified to “This study characterized the microbiota by sequencing the V3-V4 regions of prokaryotic 16S rRNA to investigate the bacterial diversity of fermented fish-entrail sauce (tai-pla or pung-pla) from five provinces in Thailand. Tai-pla samples made from seven different species of fish, three freshwater and four marine, were purchased.” Please see revised text, lines 18-21.

Line 6: "through next-generation sequencing technology" – overly general; specify "Illumina MiSeq sequencing"

Reply: We did not add these texts in the original text. We added “To do this, we used sequences from the 16S rRNA (V3-V4 regions) generated using Illumina Nova sequencing.” Lines 68-69.

Line 16: "The results showed" – tense inconsistency; use present tense: "The results show"

Reply: We did not add these texts in the original text.

Line 18: "between geographic regions and fish species" – ambiguous; specify "between different geographic regions and fish species"

Reply: We did not add these texts in the original text.

Content Issues:

  1. Vague results; no specific data on relative abundances

Reply: the scientific sound was improved, please see revised version.

  1. No mention of diversity indices or statistical analyses

Reply: We added text “Principal Coordinate Analysis visualization of beta diversity showed distinct bacterial community structures across tai-pla samples prepared with different fish species.” please see revised version, lines 29-31.

  1. No discussion of implications or conclusions

Reply: We have added already, lines 31-37.

Introduction:

Writing Issues:

Line 36: "The objectives of this study were" – tense inconsistency; use present tense: "The objectives of this study are"

Reply: We have improved already in the revised version.

Line 41: "We hypothesized" – past tense; should be present tense: "We hypothesize"

Reply: We did not add these texts in the original text.

Line 50: "and quality control" – incomplete phrase; "and quality control methods"

Reply: We did not add these texts in the original text.

Content Issues:

  1. No mention of other factors (salt concentration, pH, fermentation time)

Reply: We have added already in discussion lines 253-264.

Materials and Methods:

Writing Issues:

Line 121: "The V3-V4 hypervariable regions" – correct; note: "regions" plural

Reply: We modified thoroughly text.

Content Issues:

  1. Only 3 subsamples per sample (insufficient for technical replicates)

Reply: Three replicates per sample can be categorized in group and is the standard use for statistically significant.

  1. No details on PCR thermal cycling conditions

Reply: We have added PCR thermal cycling conditions already in the original submission, please see revised version in material and method section, lines: 114-116.

  1. No verification of amplicon size (e.g., gel electrophoresis)

Reply: We have added, please see revised version in line 117.

  1. Q20 quality threshold may be insufficient; recommend Q30

Reply: We have inserted done quality by Q30 in the original Table S1.

  1. Only 21 samples may not provide sufficient power for RDA

Reply: Redundancy Analysis (RDA) is not the aim of this study, our study aimed to study the bacterial diversity in tai-pla samples fermented from seven different types of freshwater or marine fish. We evaluated Alpha and beta diversities based on other statistical analysis i.e. the Kruskal-Wallis pairwise test and Jaccard distance.

Results:

Writing Issues:

Line 181: "Alpha diversity analysis revealed" – past tense; should be present tense: "Alpha diversity analysis reveals"

Reply: We did not add these texts in the original text.

Line 188: "between geographic regions and fish species" – ambiguous; "between different geographic regions and fish species"

Reply: We did not add these texts in the original text.

Content Issues:

  1. No mention of quality filtering parameters

Reply: Raw tags were filtered using the fastp software (Version 0.23.1) to obtain high-quality clean tags by the standard NovogeneAIT Genomics.

  1. No post-hoc test results showing which groups differ

Reply: Statistically significant differences between groups were assessed using the Kruskal-Wallis pairwise test, please see lines 141-142.

  1. Only two environmental variables considered

Reply: No environmental factor was evaluated.

  1. "Relatively low explanatory power" is an understatement (26.3% is poor)

Reply: We did not add these texts in the original text.

Discussion:

Content Issues:

  1. Vague comparison to "similar dominant phyla"

Reply: We did not add these texts in the original text.

  1. "Higher diversity" claim needs quantitative support

Reply: We did not add these texts in the original text.

  1. No mention of potential pathogenic species in these genera

Reply: We have added already in discussion section, revised manuscript, lines 339-353.

  1. Only 3-13% salt range; no optimal concentration identified

Reply: The optimal salt concentration is not the aim of this study, the aim is study the bacterial diversity in tai-pla samples fermented from seven different types of freshwater or marine fish from local markets in Thailand.

  1. Bacteroidetes acid tolerance not well-documented

Reply: We did not mention these texts in the original text.

Conclusions:

Content Issues:

  1. "First comprehensive metagenomic analysis" claim needs verification

Reply: We did not mention this analysis in the original text.

  1. "Foundation for future research" is overly optimistic

Reply: We did not mention these texts in the original text.

We suggest carefully reviewing the entire manuscript to standardize tense, eliminate ambiguity, and ensure completeness of phrases. Consider consulting a native English-speaking colleague or professional editing service to refine linguistic accuracy, which will enhance the clarity and impact of your research.

Reply: We would like to thank you very much for your kind suggestions. The revised manuscript is edited grammatically and stylistic errors by one coauthor, Professor David Blair, College of Science and Engineering, James Cook University, Townsville Qld, 4811, Australia. Scopus ID: 7202955592, the English native expert and in topic relate Genetics, Pathogens, Food borne trematodiases, Taxonomy, Evolution, Diseases, Basic science.

Reviewer 2 Report

Comments and Suggestions for Authors

This article presents an exploratory investigation. Although the sampling effort could have been stronger, since only seven samples were collected and each analyzed in triplicate, I consider the resulting information potentially valuable, and the data appear to have been analyzed appropriately. Below are several comments to improve the manuscript:

  • Abstract: The objective of the study is not clearly stated in the abstract; it only becomes evident upon reading the conclusion. It would be helpful to introduce a few sentences that provide context and clarify the purpose of the study early in the abstract.
  • Figure 1: Inferring taxonomic identification to the species level from amplicon profiling data, such as the 16S rRNA gene V3–V4 region, is not appropriate due to the limited resolution of this technique. I recommend presenting the taxonomic composition at the genus or family level instead.
  • Figure 2: The color scale does not appear to represent the relative abundance of each ASV; the presence of negative values suggests that these may reflect standardized or centered abundances (e.g., abundance relative to the mean). Additionally, some genera are labeled as pathogenic bacteria, but not all species or strains within those genera are pathogens. It would be more accurate to label them as potential pathogens.
  • Line 188: To which taxon does the ASV shared among all groups belong?
  • Line 313: Pseudomonadota does not exclusively include marine Proteobacteria. Please provide a more precise description.

Author Response

Point by point to reviewer 2

Comments and Suggestions for Authors

This article presents an exploratory investigation. Although the sampling effort could have been stronger, since only seven samples were collected and each analyzed in triplicate, I consider the resulting information potentially valuable, and the data appears to have been analyzed appropriately. Below are several comments to improve the manuscript:

Reply: We highly appreciated your supportive and constructive comments, and your comments are helpful in improving our work.

Abstract: The objective of the study is not clearly stated in the abstract; it only becomes evident upon reading the conclusion. It would be helpful to introduce a few sentences that provide context and clarify the purpose of the study early in the abstract.

Reply: We modified to “This study characterized the microbiota by sequencing the V3-V4 regions of prokaryotic 16S rRNA to investigate the bacterial diversity of fermented fish-entrail sauce (tai-pla or pung-pla) from five provinces in Thailand. Tai-pla samples made from seven different species of fish, three freshwater and four marine, were purchased.” please see lines: 18-21.

Figure 1: Inferring taxonomic identification to the species level from amplicon profiling data, such as the 16S rRNA gene V3–V4 region, is not appropriate due to the limited resolution of this technique. I recommend presenting the taxonomic composition at the genus or family level instead.

Reply: We modified as per your suggestion, please see revised Figure 1.

Figure 2: The color scale does not appear to represent the relative abundance of each ASV; the presence of negative values suggests that these may reflect standardized or centered abundances (e.g., abundance relative to the mean). Additionally, some genera are labeled as pathogenic bacteria, but not all species or strains within those genera are pathogens. It would be more accurate to label them as potential pathogens.

Reply: We modified the legend of Figure 2 “The heat map represents the top 35 genera, colors indicate the relative abundance of each ASV in each sample group category plotted by the absolute z-score.” Please see lines: 195-196. And we modified Figure 2 as per your suggestion, please see revised Figure 2.

Line 188: To which taxon does the ASV shared among all groups belong?

Reply: ASV71, k_Bacteria; p_Actinobacteriota; c_Actinomycetota; o_Propionibacteriales; f_Propionibacteriaceae; g_Cutibacterium, please see Table S4.

Line 313: Pseudomonadota does not exclusively include marine Proteobacteria. Please provide a more precise description.

Reply: To ensure clarity of this information, we removed “marine” text to “Members of other phyla represented in our samples can also have beneficial effects. For example, Pseudomonadota (proteobacteria) produces a class of bioactive peptides through nonribosomal pathways, which have demonstrated potent antibacterial and/or antifungal properties [30]. Members of the phylum Actinomycetota, Family Micrococcaceae, play a key role in the synthesis of bioactive compounds exhibiting a wide spectrum of therapeutic and antimicrobial properties [31].” please see line 332.

Finally, again, we would like to express our sincere appreciation for your valuable comments and suggestions, which have greatly helped us improve the manuscript. We hope the revised version meets your expectations.